# Molecular Cloning of a TCHQD Class Glutathione S-Transferase and *GST* Function in Response to GABA Induction of Melon Seedlings under Root Hypoxic Stress

Jingrui Li [†], Chunyan Wang [†], Xiaolei Wu, Binbin Gong, Guiyun Lü and Hongbo Gao *

College of Horticulture, Hebei Collaborative Innovation Center of Green and Efficient Vegetable Industry, Key Laboratory of North China Water-Saving Irrigation Engineering, Ministry of Agriculture and Rural Affairs, Hebei Agricultural University, Baoding 071001, China; yyljr@hebau.edu.cn (J.L.); wangchunyan321@126.com (C.W.); yywxl@hebau.edu.cn (X.W.); yygbb@hebau.edu.cn (B.G.); yylgy@hebau.edu.cn (G.L.)
* Correspondence: hongbogao@hebau.edu.cn
† These authors contributed equally to this work.

**Abstract:** Glutathione-S-transferase (GST), a versatile enzyme that occurs widely in plants, plays a key role in plant resistance to environmental stresses. Previous results have demonstrated that GST proteins are involved in alleviating root hypoxic injury caused by gamma-aminobutyric acid (GABA); however, the induction mechanism of the *GST* gene in the melon under root hypoxic stress and its functional mechanisms remain unclear. In this study, based on gene cloning and bioinformatics analysis, *GST* gene expression and activity and glutathione (GSH) content were assessed under root hypoxic and normoxic conditions with or without GABA. The results showed that the *CmGST* locus includes an 804 bp gene sequence that encodes 267 amino acids. The sequence was highly similar to those of other plant TCHQD *GSTs*, and the highest value (94%) corresponded to *Cucumis sativus*. Real-time PCR results showed that the *CmGST* gene was induced by root hypoxic stress and GABA, and this induction was accompanied by increased GST activity and GSH content. Root hypoxic stress significantly upregulated *CmGST* expression in melon roots (0.5–6 d), stems, and leaves (0.5–4 d), and GST activity and GSH content were also significantly increased. Exogenous GABA treatment upregulated *CmGST* gene expression, GST activity, and GSH content, particularly under root hypoxic conditions. As a result, *CmGST* expression in GABA-treated roots and leaves at 0.5–4 d and stems at 0.5–6 d was significantly higher than that under root hypoxic stress alone. This study provides evidence that the TCHQD *CmGST* may play a vital role in how GABA increases melon hypoxia tolerance by upregulating gene expression and improving metabolism.

**Keywords:** enzyme activity; gamma-aminobutyric acid; gene clone; *GST*; TCHQD; gene expression; root hypoxic condition



## 1. Introduction

Glutathione-S-transferase (GST) is a multifunctional enzyme that exists widely in plants. It plays an important role in enhancing plant resistance by catalyzing glutathione (GSH) to form conjugates with hydrophobic and electrophilic compounds, which are sent to vacuoles or plastids [1]. Genome-wide analysis showed that *GST* genes were widely distributed in plants, belonged to a multigene family, and were highly conserved in structure and function [2]. Plant GSTs are classified into fourteen distinct classes: zeta (Z), theta (T), iota (I), lambda (L), tau (U), phi (F), dehydroascorbate reductase (DHAR), elongation factor 1Bγ (EF1Bγ), glutathionyl hydroquinone reductase (GHR), hemerythrin, microsomal prostaglandin esynthase type 2 (mPGES-2), tetrachloro-hydroquinone dehalogenase (TCHQD), ure2p, and metaxin [3]. The functions of different subgroups of *GST* genes differed. For example, tau and phi GSTs are involved in xenobiotic metabolism [4,5]. However, little is known about the roles of TCHQD *GST*.

Further studies have shown that *GST* gene expression and metabolic changes are closely related to various abiotic stresses in plants. For example, cold stress can induce a significant increase in *GST* gene expression in cold-tolerant pumpkin [6], while hypoxic stress can induce *GST* gene expression in rice [7]. Drought, waterlogging, salt, and other stresses can significantly increase *GST* gene expression in different plants [8,9]. GSH reacts with membrane lipid peroxides to catalyze the removal of excessive reactive oxygen species (ROS) in vivo, thus reducing the damage to the plant cell membrane structure caused by stress [10]. When the *OsGSTL2* gene was transferred into *Arabidopsis thaliana*, the tolerance of *Arabidopsis* to heavy metals was enhanced, and its tolerance to low temperature, osmotic stress, and salt stress was also improved [8]. However, *GST* gene expression patterns differ among stress.

Hypoxia is caused by flooding and soil hardening, which leads to decreased crop yields and economic losses. Melon (*Cucumis melo*) is a typical hypoxia-sensitive plant species, and excessive irrigation or soil compaction leads to hypoxic stress, which results in excessive reactive oxygen species (ROS) in melon cells, interferes with the normal physiological metabolism of cells, and ultimately reduces the yield and quality of melon [11]. Previous studies have shown that enhanced gamma-aminobutyric acid (GABA) accumulation could effectively enhance the tolerance of wheat plants to hypoxic stress [12]. In addition, our previous studies used two-dimensional electrophoresis and mass spectrometry to show that GABA can induce the production of 13 specific proteins, including GST, with significantly different expression levels under hypoxic stress in melon [13]. It is speculated that GST plays an important role in the hypoxia-tolerance response of melons induced by GABA. However, it is still unknown whether TCHQD *GST* participates in the regulation of plant hypoxia tolerance via GABA. There are few reports about the TCHQD *GST* gene, and the TCHQD *GST* gene in melons has not been cloned. The expression pattern of the TCHQD *GST* gene in plants under root hypoxic stress is still unclear.

Therefore, the full-length sequence of the coding DNA sequence (CDS) region of the TCHQD *GST* gene was cloned, and the function of the gene was predicted by bioinformatics. Combined with the study of the temporal and spatial expression patterns of the TCHQD *GST* gene, GST enzyme activity, and GSH substrate metabolism under root hypoxia stress induced by GABA, the mechanism of *GST* was clarified. The study provides evidence for the regulatory role of GST in root hypoxia stress induced by GABA and improves the hypoxia tolerance of melons. It also provides a reference for the molecular breeding of melons.

## 2. Materials and Methods

### 2.1. Materials

The experiment was completed in an environmentally controlled greenhouse at Hebei Agricultural University. The hypoxia-sensitive melon variety Xiyu No. 1 was used as the experimental material.

### 2.2. Experimental Design

Melon seeds of uniform size were soaked in water at 55 °C and then germinated in a thermostat ($26 \pm 1$ °C). Seeds with 1–2 mm radicles were sown in quartz sand and cultured at 27 °C~30 °C/16 °C~18 °C (day/night). The cotyledons were flattened and irrigated with half-strength Hoagland nutrient solution every 2–3 d. When the seedlings were at the three-true-leaf stage, uniform seedlings were selected and transplanted into plastic troughs (60 cm × 40 cm × 20 cm) filled with a full-strength Hoagland solution. The seedlings were cultivated in an environmentally controlled greenhouse at 25–30 °C/15–18 °C day/night temperature, 75–85% relative humidity, and 800 $\mu mol \cdot m^{-2} \cdot s^{-1}$ photosynthetic photon flux density. The dissolved oxygen concentration (DO) of the nutrient solution was maintained at $8.0 \pm 0.2$ $mg \cdot L^{-1}$ by an air pump. At the three-leaf stage, the seedlings were divided into four groups and subjected to the following treatments:

(1) Normoxia: the seedlings were cultured with normal full-strength Hoagland solution into which air was pumped, and an dissolved oxygen analyzer (Pisco DO500, Berlin, Germany) was used to maintain nutrient solution DO at $8.0 \pm 0.2$ mg·L$^{-1}$.

(2) Normoxia + GABA: GABA (5 mmol·L$^{-1}$) was added to the normal full-strength Hoagland solution (the GABA concentration was the best concentration selected in the pre-experiment), and the DO was maintained at $8.0 \pm 0.2$ mg·L$^{-1}$.

(3) Hypoxia: the DO was maintained at $2 \pm 0.2$ mg·L$^{-1}$ by adding N$_2$ to the full-strength Hoagland solution.

(4) Hypoxia + GABA: GABA (5 mmol·L$^{-1}$) was added to the nutrient solution based on the hypoxia treatment.

During seedling culture, the growth of the melon seedlings under root hypoxic stress was severely inhibited compared with that of the control seedlings. Our previous study showed exogenous GABA treatment effectively alleviated the inhibition of seedling growth caused by root hypoxic stress.

Four days after treatment, the second fully unfolded leaf was cut at the growth point and used for *CmGST* gene cloning and sequence structure analysis. The middle root (root), the first-second segment (stem), and the second fully unfolded leaf (leaf) at the growth point were cut 0, 0.5, 1, 2, 3, 4, and 6 d after treatment. The *GST* gene expression, GST activity, and GSH content in the roots, stems, and leaves of the seedlings were measured 0, 2, 4, 6, and 8 d after treatment (Figure 1). Three plants were taken from each treatment, and measurements were repeated three times.

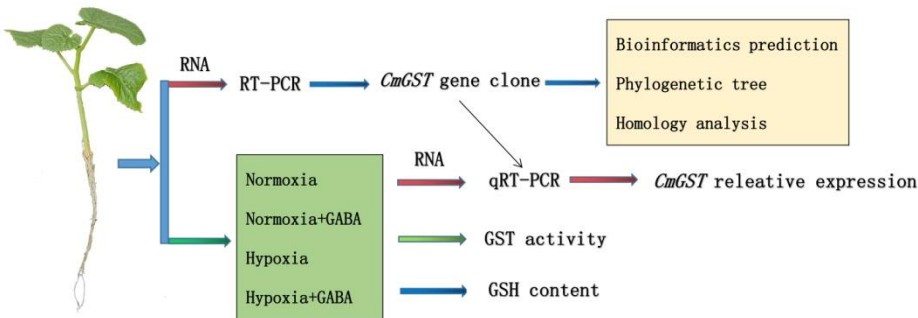

**Figure 1.** Technology roadmap.

### 2.3. Extraction of Total RNA from Melon Seedlings and Synthesis of First-Strand cDNA

Total RNA was extracted from roots, stems, and leaves using the EASYEx PLUS Plant RNA Extraction Kit (TianGen, Beijing, China). Detection of RNA integrity by agarose gel electrophoresis. The TaKaRa First-Strand Reverse Transcriptase Kit (TransGen, Beijing, China) was used to synthesize the first-strand cDNA using total RNA as a template. Then, 0.5 µg RNA was added to the 20 µL solution for the synthesis of cDNA.

### 2.4. CmGST Gene Cloning

To obtain the *CmGST* gene coding sequence, using the conserved sequence of the *C. sativus* TCHQD *GST* gene as a reference, Primer Premier 5.0 was used to design specific amplification primers: GCTGGATGTTATCTGGGTCTC and ATTCAACCTCCAATCCTCTGT.

Using melon leaf RNA as a template, *GST* amplification was carried out by RT-PCR. The amplification reaction system (25 µL) was as follows: 2.5 µL of 10 × PCR buffer, 4 µL of a dNTP mixture (2.5 mmol·L$^{-1}$), 0.5 µL of forward and reverse primers (10 µmol·L$^{-1}$), 0.25 µL of Taq DNA polymerase (5 U·µL$^{-1}$), 1.25 µL of template DNA (300 ng·mL$^{-1}$), and 16 µL of RNase-ddH$_2$O. The PCR amplification procedure was as follows: predenaturation at 94 °C for 30 s; 40 cycles of denaturation at 94 °C for 30 s, annealing at 55 °C for 30 s, and extension at 72 °C for 30 s; and a final extension at 72 °C. The PCR products were detected by 1% agarose gel electrophoresis, and gene sequencing was performed. A 1000 bp band was purified from agarose gel using TIANgel Midi Purification Kit (TianGen, Beijing, China) and sequenced by Huada Gene Technology Co. Ltd (Shenzhen, China).

## 2.5. Bioinformatic Analysis of CmGST

To predict GST protein function, ORF Finder (https://www.ncbi.nlm.nih.gov/orffinder/, 1 May 2022) was used to predict the open reading frame; BLAST (https://blast.ncbi.nlm.nih.gov/Blast.cgi, 1 May 2022) was used for homology analysis; the GST protein sequence of melon was analyzed by DNAMAN 9.0; the phylogenetic tree was constructed by MEGA 11. ProtParam (https://web.expasy.org/protparam/, 1 May 2022) was used to predict the physical and chemical properties of melon GST protein; SignalP 4.1 Server (https://services.healthtech.dtu.dk/service.php?SignalP, 1 May 2022) was used to predict the signal peptide-encoding protein; InterProScan (https://www.ebi.ac.uk/interpro/, 1 May 2022) was used to predict the conserved domain of the gene; ProtScale (https://www.expasy.org/resources/protscale, 1 May 2022) analysis was used to predict the hydrophobicity/hydrophilicity of the amino acid sequence. PSORT II (http://psort.hgc.jp/, 1 May 2022) predictions were used to predict subcellular localization, and GOR IV was used to predict the secondary structure of the protein.

## 2.6. Real-Time Fluorescence Quantitative PCR

In order to study the response of the *GST* gene of exogenous GABA on melon seedlings under root hypoxia stress, we measured the relative expression of *CmGST* gene at different times and in different parts of the plants. Specific primers were designed by Primer Premier 5.0 according to the obtained *GST* sequence: GGATAGCCAGAAGGTGAGAC and AGTTTAGCACTTGGGTTGAT. The *actin* gene (FJ763186) registered in GenBank was used as an internal reference gene; the primers for this gene were CCGAAGCAAAG-GAAGA and TTGTCCGACCACTGGCATAGAG. The reaction system consisted of 12.5 μL of 2 × SuperReal PreMix (TransGen, Beijing, China), 0.75 μL of forward and reverse primers (10 μmol·L$^{-1}$), and 2 μL of cDNA, with a final volume of 25 μL, achieved by the addition of ddH$_2$O. The reaction procedure was as follows: 95 °C for 15 s, followed by 40 cycles of 95 °C for 15 s, 58 °C for 20 s, and 72 °C for 30 s. Each sample was treated three times, and the results were calculated by the $2^{-1\Delta\Delta ct}$ method.

## 2.7. GST Activity and GSH Content Analysis

In order to study whether exogenous GABA scavenges GSH through the GST enzyme of melon seedlings under root hypoxia stress, GST activity and GSH content were measured at different times and in different parts of the plants. For each treatment, 0.2 g of each plant tissue was weighed, and 1.8 mL of 0.1 mol·L$^{-1}$ PBS buffer solution, pH 7.4, was added. After the tissue was ground, the homogenate was centrifuged at 12,000 rpm for 20 min. The supernatant was collected, and the GST activity was determined by using the plant GST ELISA Kit (Trust Specialty Zeal biological Trade Co., Ltd. Massachusetts, USA).

The root, stem, and leaf tissues of melon treated with different treatments (0.5 g) were weighed, ground into a homogenate with 5% trichloroacetic acid (TCA), and centrifuged at 12,000 rpm. The GSH content was determined by the 5,5-dithio-p-nitrobenzoic acid (DNTB) colorimetric method. The absorbance at 412 nm was determined by an ultraviolet spectrophotometer.

## 2.8. Statistical Analysis

Microsoft Office Excel software 2007 was used for data processing and drawing. Data significance analysis was performed by Duncan multiple comparison method ($p < 0.05$) of SAS 8.1 (SAS Institute, Cary, NC, USA) software.

## 3. Results

### 3.1. Cloning and Sequence Analysis of CmGST

Approximately 1000 bp of the specific target fragment was amplified by RT–PCR based on the designed specific amplification primers. The PCR products were sent to Huada Gene Technology Co. Ltd (Shenzhen, China). for purification and sequencing. After sequencing, a 1076 bp sequence was obtained by DNAMAN 9.0 (Figure S1). NCBI ORF

Finder (https://www.ncbi.nlm.nih.gov/orffinder/, 1 May 2022) was used to predict the open reading frame. The results showed that the length of the complete open reading frame of the *CmGST* sequence was 804 bp, encoding 267 amino acids. The obtained sequence was predicted to be the *CmGST* gene (Figure S2).

The molecular formula of the CmGST protein is $C_{1431}H_{2239}N_{381}O_{397}S_8$. The theoretical isoelectric point and molecular weight of the CmGST protein are predicted to be 9.26 and 31,389.3, respectively. Hydrophobic/hydrophilic prediction showed that the protein is hydrophilic (Figure S3). The protein was predicted to distribute mainly in cell microbodies (61.1%) and less so in mitochondria (10%) and chloroplasts (10%). The CmGST proteins are predicted to consist of α-helices, β-folds, and irregular curls, accounting for 43.1%, 23.6%, and 33.3% of the total protein structure, respectively (Figure S4). The amino acid sequence of the protein does not contain a signal peptide digestion site, signal peptide, or transmembrane helix (Figure S5).

### 3.2. Molecular Evolution Analysis and Amino Acid Sequence Alignment

To study the relationship between the amino acid sequence of the *CmGST* gene and the *GST* genes of other classes, a phylogenetic tree of *CmGST* from 33 GSTs from various plant species was constructed by using the proximity method in MEGA 6.0 (Figure 2). This phylogenetic tree showed that Phi, Theta, EF1Bγ, Zeta, Tau, DHAR, lambda, and GHR classes of the plant GST were grouped with strong bootstrap support. CmGST TCHQD was also placed within the TCHQD class with strong bootstrap support. Therefore, based on the amino acid sequence similarity and phylogenetic result, *CmGST* TCHQD could be classified as a TCHQD class GST.

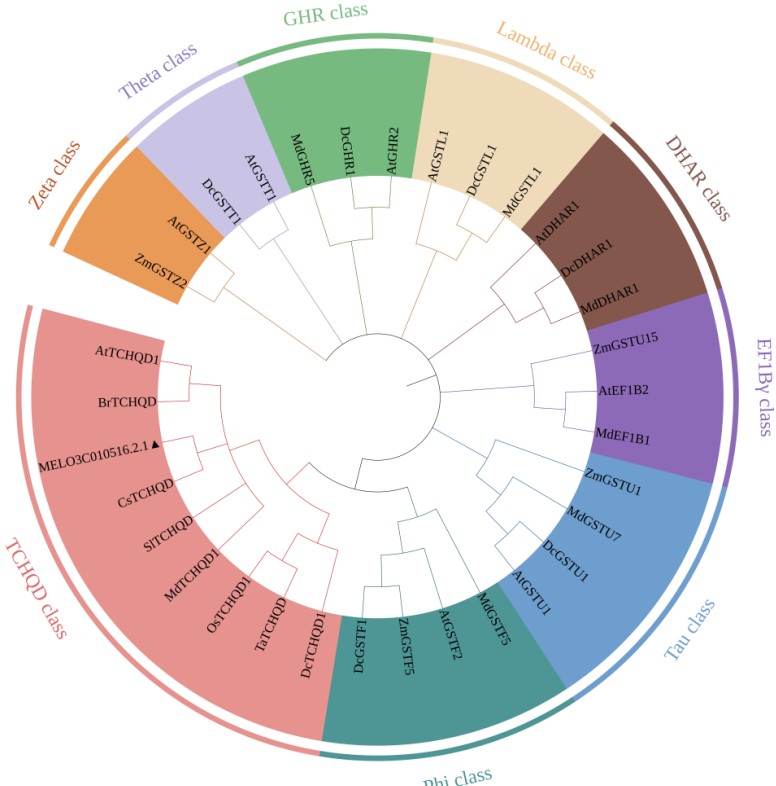

**Figure 2.** Phylogenetic relationships between *MELO3C010516.2.1* (*CmGST* TCHQD, highlighted by black triangle) and other plant GST classes. The phylogenetic tree was constructed with the MEGA11 program using maximum likelihood method, and bootstrap analysis was carried out with 1000 replications. The affiliations of the plant GST sequences used in the tree reconstruction are the following: *MELO3C010516.2.1*; *C. melo* GST TCHQD, *CsTCHQD*; *C. sativus* GST TCHQD (Cucsa.018410.1),

*SlTCHQD*; *Lycopersicon esculentum* GST TCHQD (Solyc04g057890.2), *AtTCHQD1*; *Arabidopsis thaliana* GST TCHQD (NP_177853.1), *BrTCHQD*; *Brassica rapa* L. ssp. *pekinensis* GST TCHQD (Brara.G03507.1.p), *MdTCHQD1*; *Malus pumila* GST TCHQD (MD15G1133600), *DcTCHQD1*; *Dracaena draco* GST TCHQD (KU565021), *OsTCHQD1*; *Oryza sativa* GST TCHQD (LOC_Os04g35560.1), *TaTCHQD*; *Triticum aestivum* GST TCHQD (Traes_2BL_32FE5AEF9.2), *MdGSTF5*; *Malus pumila* GST Phi (MD17G1133600), *AtGSTF2*; *Arabidopsis thaliana* GST Phi (NP_192161.1), *DcGSTF1*; *Dracaena draco* GST Phi (KU565000), *ZmGSTF5*; *Zea mays* GST Phi (NP_001105111.2), *AtGSTT1*; *Arabidopsis thaliana* GST Theta (NP_198937.1), *DcGSTT1*; *Dracaena draco* GST Theta (KU565024), *ZmGSTU15*; *Zea mays* L. GST EF1Bγ (NP_001131533.2), *MdEF1B1*; *Malus pumila* GST EF1Bγ (MD08G1244100), *AtEF1B2*; *Arabidopsis thaliana* (L.) Heynh. GST EF1Bγ (NP_176084.1), *AtGSTZ1*; *Arabidopsis thaliana* GST Zeta (NP_178344.1), *ZmGSTZ2*; *Zea mays* GST Zeta (XP_008661994.1), *ZmGSTU1*; *Zea mays* GST Tau (NP_001104989.1), *MdGSTU7*; *Malus pumila* GST Tau (MD10G1196300), *AtGSTU1*; *Arabidopsis thaliana* GST Tau (XP_020885836.1), *DcGSTU1*; *Dracaena draco* GST Tau (KU565010), *MdDHAR1*; *Malus pumila* GST DHAR (MD17G1260600), *DcDHAR1*; *Dracaena draco* GST DHAR (KU565019), *AtDHAR1*; *Arabidopsis thaliana* GST DHAR (NP_173387.1), *AtGSTL1*; *Arabidopsis thaliana* GST Lambda (CAB86032.1), *MdGSTL1*; *Malus pumila* GST Lambda (MD12G1129400), *DcGSTL1*; *Dracaena draco* (L.) L. GST Lambda (KU565022), *MdGHR5*; *Malus pumila* GST GHR (MD05G1251900), *AtGHR2*; *Arabidopsis thaliana* GST GHR (NP_199312.1), *DcGHR1*; *Dracaena draco* GST GHR (KU565023).

The amino acid sequence of the *CmGST* gene was compared with the sequences in the NCBI database via BLAST. Eight CmGST amino acid sequences of other plants with relatively high homology were selected from the comparison results (Figure 3). DNAMAN analysis showed that the amino acid sequence encoded by the *CmGST* gene had a high homology, 94%, to a gene in cucumber (*C. sativus*, XP_004142639.1), followed by those in tomato (72%; *Lycopersicon esculentum*, Solyc04g057890.2) and apple (70%; *Malus pumila*, MD15G1133600). Therefore, *CmGST* was presumed to be highly conserved.

### 3.3. CmGST Relative Expression under Exogenous GABA Treatment Regulates Root Hypoxic Stress

During the entire experiment, the expression of *CmGST* first increased and then decreased under normoxia control, hypoxic stress, normoxia + GABA, and hypoxia + GABA treatments, and there were significant spatial and temporal differences (Figure 4). Compared with the control, the *CmGST* gene was significantly upregulated in roots 0.5–6 d after treatment and in stems and leaves after 0.5–4 d under hypoxic stress, with increases of 1.93–3.73 times, 1.78–5.08 times, and 3.61–6.21 times, respectively. Exogenous GABA treatment further promoted the upregulation of *CmGST* gene expression under hypoxic stress. The expression of *CmGST* in roots and leaves 0.5–4 d after treatment and in stems after 0.5–6 d was significantly higher than that in the hypoxic stress treatment. After treatment for 0.5 d, the increases were 49.6%, 117.1% and 262.2%, respectively. Under normoxia control, exogenous GABA significantly promoted the upregulation of *CmGST* expression, but this expression was significantly lower than that under the hypoxic stress + GABA treatment throughout the entire treatment duration (0.5–6 d).

### 3.4. GST Activity under Exogenous GABA Treatment Regulates Root Hypoxic Stress

The changes in GST activity in the roots, stems, and leaves were similar under different treatments, but the GST activity in the roots was higher than that in the stems and leaves (Figure 5). Compared with the control treatment, the normoxia + GABA treatment and hypoxic stress treatment significantly increased the GST activity in the roots, stems, and leaves of melon seedlings, but the GST activity under the hypoxic stress treatment was high throughout the treatment duration, and the GST activity in the roots, stems, and leaves was significantly higher than that under the control treatment. Under the hypoxic stress + GABA treatment, the GST activity in the roots, stems, and leaves further increased, and after 2–6 d of treatment, the activity was significantly higher than that under hypoxic stress.

The GST activity in the roots, stems, and leaves increased by 21.3%, 21.6%, and 16.5%, respectively, after 4 d of hypoxia + GABA treatment.

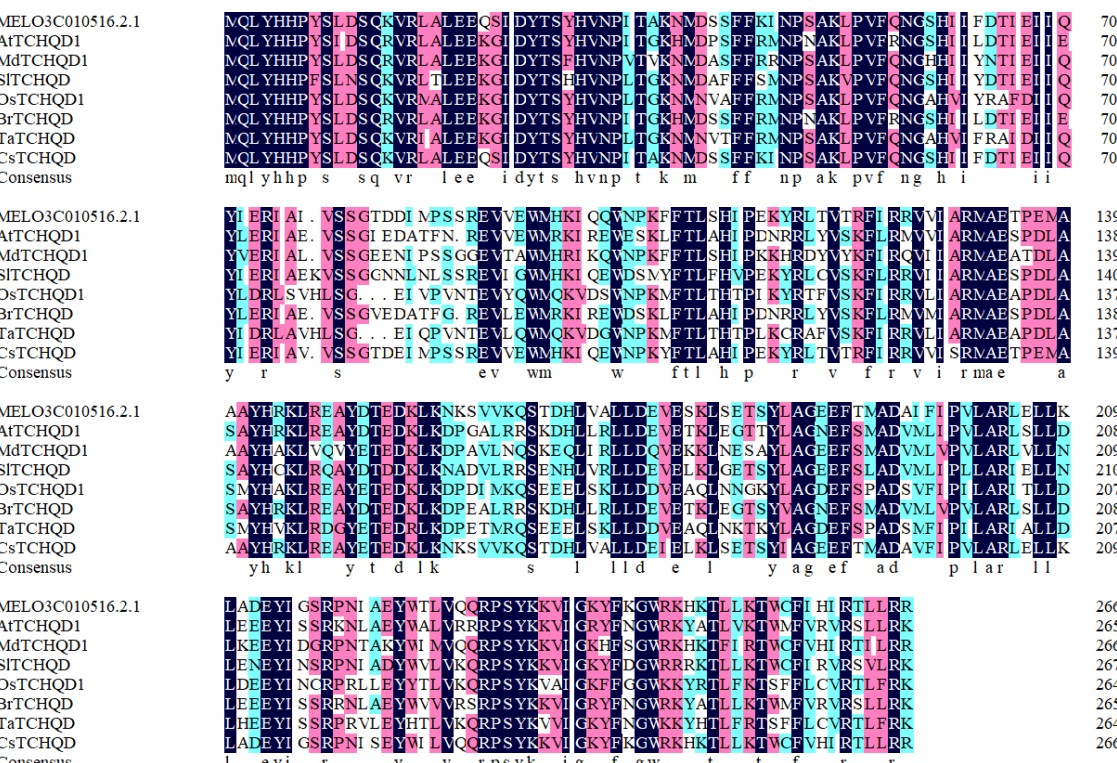

**Figure 3.** Multiple alignments of the amino acid sequences of *CmGST* and those of other species. Conserved residues in all plant TCHQD GSTs are indicated below. *MELO3C010516.2.1*; *C. melo* GST TCHQD, *AtTCHQD1*; *Arabidopsis thaliana* GST TCHQD (NP_177853.1), *MdTCHQD1*; *Malus pumila* GST TCHQD (MD15G1133600), *SlTCHQD*; *Lycopersicon esculentum* Miller GST TCHQD (Solyc04g057890.2), *OsTCHQD1*; *Oryza sativa* GST TCHQD (LOC_Os04g35560.1), *BrTCHQD*; *Brassica rapa* GST TCHQD (Brara.G03507.1.p), *TaTCHQD*; *Triticum aestivum* GST TCHQD (Traes_2BL_32FE5AEF9.2), *CsTCHQD*; *C. sativus* GST TCHQD (Cucsa.018410.1) are included in the alignment.

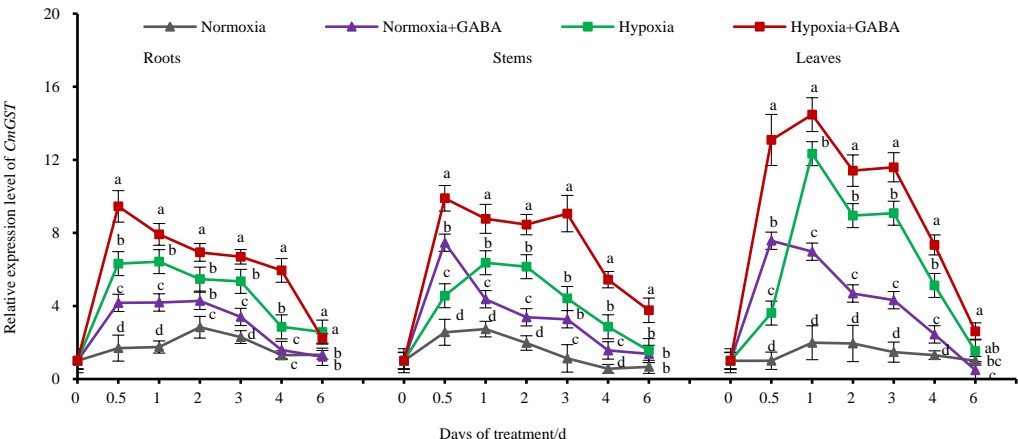

**Figure 4.** Effects of exogenous GABA on the relative expression of *CmGST* in the roots, stems, and leaves of melon seedlings under root hypoxic stress for 0–6 d. Different letters a–d indicate significant difference ($p < 0.05$).

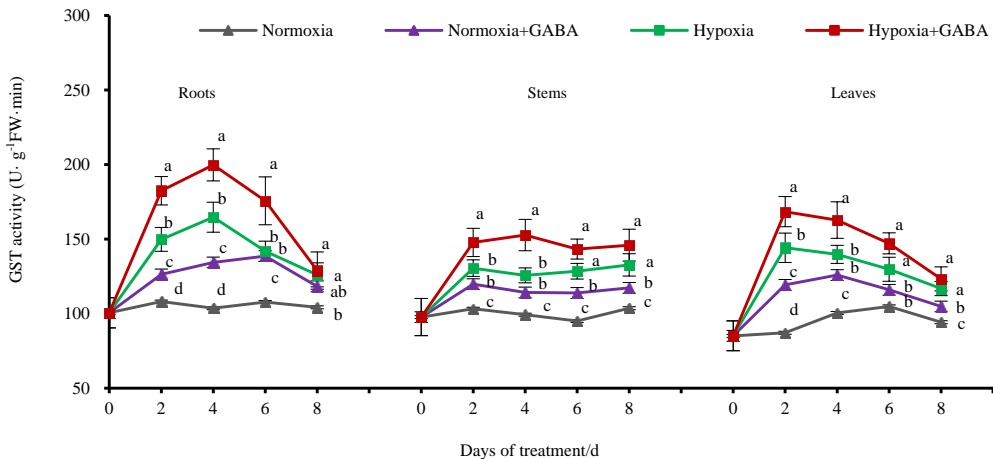

**Figure 5.** Effects of exogenous GABA on GST activity in the roots, stems, and leaves of melon seedlings under root hypoxic stress for 0, 2, 4, 6, and 8 d. Different letters a–d indicate significant difference ($p < 0.05$).

### 3.5. GSH Content under Exogenous GABA Treatment Regulates Root Hypoxic Stress

The changes in the GSH content in the roots, stems, and leaves of melon seedlings were consistent from 0–8 d under different treatments (Figure 6). The hypoxic stress + GABA treatment produced the highest GSH content, followed by the hypoxic stress treatment and normoxia + GABA treatment, while the control treatment produced the lowest GSH content; moreover, the GSH content in the leaves was the highest, followed by that in the stem and then that in the root. The root GSH content under the hypoxic stress + GABA treatment was 18.7–24.0% higher than that under the hypoxic stress treatment at 2–4 d, while the GSH content in the leaf under the hypoxic stress + GABA treatment was significantly higher than that under the hypoxic stress treatment at 2–6 d. The GST activity and GSH content in the belowground parts were always higher than those in the aboveground parts.

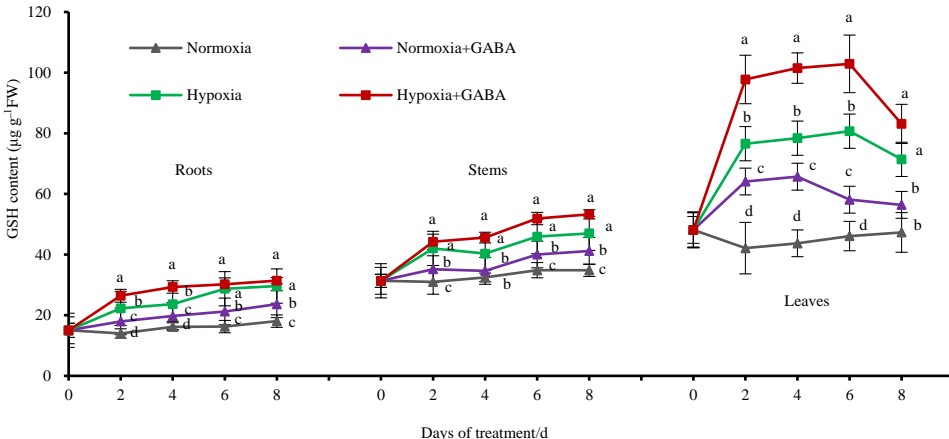

**Figure 6.** Effects of exogenous GABA on GSH contents in roots, stems, and leaves of melon seedlings under root hypoxic stress for 0, 2, 4, 6, and 8 d. Different letters a–d indicate significant difference ($p < 0.05$).

### 4. Discussion

Under abiotic stresses, aldehydes and ROS accumulate in the plant; plants produce specialized metabolites such as regulatory signals (e.g., phytohormones) to structural polymers (e.g., lignin) and activate a protective enzyme system to remove harmful substances protecting plants from stress [14]. Aldehyde dehydrogenase (ALDH) detoxifies aldehydes generated by ethanol fermentation, lipid peroxidation, and environmental stresses [15]. Antioxidant enzyme system, including SOD, CAT, POD, and GST, plays an important

role in scavenging ROS. GST can catalyze the reaction between GSH and hydrophobic and electrophilic compounds, thus promoting the metabolism of toxic heterologous substances and oxidation products and regionalized isolation or elimination [16]. It can play a role in plant primary metabolism, secondary metabolism, stress response, and cell signal transduction, thus affecting plant growth and development. To date, *GST* genes have been identified in barley [17], Chinese cabbage [18], and pear [19]. Further studies have shown that *GST* in plants is encoded by a multigene family. According to GST protein homology and genome structure, *GST* can be divided into 14 groups that perform functions in primary and secondary metabolism, stress tolerance, cell signal transduction, and so on [20]. In the study, we cloned the *CmGST* gene by RT-PCR; it contains an 804 bp open reading frame and encodes 267 amino acid sequences. The CmGST protein is composed of α-helices, β-folds, and irregular curls and is distributed mainly in cell microbodies. There are functional regions at both ends of the GST protein amino acid sequence, but no signal peptide exists. Further analysis showed that the *GST* gene sequence of melons was the TCHQD class. The TCHQD GST was found in Arabidopsis, cucumber, tomato, and so on, and the highest homology with melon TCHQD GST was 94% in cucumber. This showed that GSTs are highly conserved in evolution.

GABA is widely distributed throughout the biological world, and environmental stresses increase GABA accumulation [21]. GABA is considered to be a putative signaling molecule that might have acted in the molecular physiology of the earliest land plants [22]. In creeping bentgrass, GABA-induced heat tolerance involves the enhancement of photosynthesis and ascorbate-glutathione cycle [23]. Our study showed that exogenous GABA induced GST protein expression in melons under root hypoxic stress [13]; exogenous GABA also increased white clover *GST* gene transcript levels [24]. GABA enhances tolerance to phenanthrene stress via a GSH-dependent system of antioxidant defense in cucumber, and GSH content is enhanced [25]. *GST* expression in salicylic acid-treated tomato was upregulated under salt stress, which promoted GST activity [26]. The catalytic functions of GSTs include GSH conjugation in the metabolic detoxification of herbicides and natural products. GSTs can also catalyze GSH-dependent peroxidase reactions that scavenge toxic organic hydroperoxides and protect against oxidative damage [9]. Therefore, the increase in *GST* gene expression and enzyme activity is closely related to the response to various stresses, but the effects of different stress conditions in different varieties are also quite different. Compared to untreated control, pepper GST activity was enhanced significantly under salinity, heat, and oxidative stresses, and the increase in GST activity was mainly affected by the upregulation of *GST* gene expression [27]. Compared with the WT, transgenic *MruGSTU39* lines showed significantly enhanced *GST* relative expression and thus significantly enhanced GST activity [28]. At the same time, the expression of *GST* genes in different crops was induced by different stresses and times. For example, the expression of *osgstu 4* and *osgstu 3* genes in rice increased significantly after 2, 6, and 12 h of hypoxia treatment [8]. Tomato *LeGSTU2* reached its highest transcript level at 6 h after salt or osmotic treatment. Under heat stress, *LeGSTU2* transcripts were the highest at 3 h [29]. Further studies showed that changes in *GST* gene expression and metabolism in different plants under stress could be affected by exogenous substances. The maize *GST* gene was upregulated by ABA, IAA, and GA [30]. The results showed that root hypoxic stress and GABA could synergistically induce *CmGST* gene expression and that there were significant spatial and temporal differences. Roots (0.5–6 d), stems, and leaves (0.5–4 d) showed significant increases in *CmGST* gene expression under hypoxia treatment, and exogenous GABA treatment further promoted *CmGST* gene expression at 0.5–4 d of hypoxic stress, which showed that *CmGST* gene expression significantly increased in roots and leaves (0.5–4 d) and stems (0.5–6 d) under hypoxia treatment. The changes in GST activity and GSH content in melon were similar to those for *CmGST*, but the changes in GST activity and GSH content were small, indicating that exogenous GABA could be transported to stems and leaves through vascular bundles after rapid absorption by melon roots. GST activity in chloroplasts was activated by inducing *CmGST* gene expression and catalyzing the binding

reaction between GSH and various electrophilic exogenous chemicals. Thus, the hypoxia tolerance of melon seedlings was enhanced while maintaining a higher photosynthetic rate [11].

Under the same conditions, *GST* expression was different in different parts of the same plant, and the GST activity and GSH content were also different. In *Brassica napus*, 2 *GST* genes were expressed in all 21 tissues, whereas 9 *GST* genes were not expressed in all tissues; some *GST* genes were specifically expressed in different tissues [31]. *SbGSTU7* and *SbGSTU12*, showed a much higher expression level in the root tissues than that in the primary leaf, mature leaf, phloem, bud, and flower tissues [32]. GST activities were increased in pumpkin (3–12 d), with 22.5–47.3% activation under treatment with perfluorooctane sulfonamide [33]. Under antimony stress, GSH content in wheat roots was increased more than that in shoots, demonstrating that GSH played a limited role in wheat shoots [34]. The GST activity increased significantly in both leaves and roots in *Biscutella auriculata* under cadmium stress. GSH content in leaves was significantly higher than those in roots, while GST activity in roots was significantly higher than those in leaves [35]. The results showed that there were significant differences in *CmGST* gene expression, GST activity, and GSH content in different melon tissues under the same treatment. The order of *CmGST* gene expression was leaf > stem > root, the order of GST activity was root > leaf > stem, and the order of GSH content was stem > leaf > root. These results indicated that GST activity was induced by *CmGST* gene expression, but this effect was more closely related to hypoxia and GABA treatment and to the different plant parts. There were differences in response between different parts, which may be closely related to the expression of the *CmGST* gene in different parts of the melon. Expression of GST was higher in leaves than in roots and stem, but the activity of GST was greater in roots rather than in leaves. *GST* gene expression is the main factor affecting GST activity, but the process of protein product formation is very complex, and there may be other factors affecting protein product formation. In addition, the time of gene expression takes precedence over the time of protein formation, which is also related to different tissues. The changes in GST activity and GSH content in different melon tissues were negatively correlated. It is speculated that the higher GST activity in roots can promptly catalyze the GSH reaction, resulting in the lowest GSH content in roots, while the stems and leaves may be affected by chlorophyll or anthocyanin. Under the influence of isometabolism, lower GST activity resulted in a higher GSH content, which resulted in the difference in hypoxia tolerance of different tissues.

## 5. Conclusions

In this study, we cloned the TCHQD *GST* gene, which includes an 804 bp gene sequence that encodes 267 amino acids. Bioinformatics analysis showed that the sequence was highly similar to those of other plant *GSTs*, and the highest value (94%) corresponded to *C. sativus*. *CmGST* expression, GST activity, and GSH content were organ-specific. The *CmGST* gene was induced by root hypoxic stress and GABA, and this induction was accompanied by increased GST activity and GSH content. *CmGST* may play a vital role in how GABA increases melon hypoxia tolerance by upregulating gene expression and improving metabolism. The study provides evidence for the regulatory role of GST in root hypoxia stress induced by GABA and improves the hypoxia tolerance of melons. It also provides a reference for the molecular breeding of melons.

**Supplementary Materials:** The following supporting information can be downloaded at: https://www.mdpi.com/article/10.3390/horticulturae8050446/s1. Figure S1: RT-PCR amplification of *CmGST*; Figure S2: Complete nucleotide sequence of *CmGST*; Figure S3: The secondary structure of CmGST predicted; Figure S4: Predicted hydrophobicity / hydrophilicity of CmGST; Figure S5: Predicted signal peptide of CmGST.

**Author Contributions:** Project administration, H.G.; methodology, J.L. and C.W.; formal analysis, B.G. and X.W.; writing—original draft preparation, J.L.; writing—review and editing, H.G. and G.L. All authors have read and agreed to the published version of the manuscript.

**Funding:** The research was funded by the Natural Science Foundation of Hebei (C2014204074), Modern Agricultural Industry Technology System Facilities Vegetable Innovation Team of Hebei (HBCT2021030213) and "Three-Three-Three Talent Project" Funding for Talent Training of Hebei (A201901044).

**Institutional Review Board Statement:** Not applicable.

**Informed Consent Statement:** Not applicable.

**Data Availability Statement:** Not applicable.

**Acknowledgments:** We thank all of our colleagues in our laboratory for providing useful technical support.

**Conflicts of Interest:** The authors declare no conflict of interest.

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
