# Peer review of "Molecular Cloning of a TCHQD Class Glutathione S-Transferase and GST Function in Response to GABA Induction of Melon Seedlings under Root Hypoxic Stress"

_horticulturae, doi:10.3390/horticulturae8050446_

Round 1

Reviewer 1 Report

I am commenting on the article "Molecular Cloning of a TCHQD Class Glutathione S-Transferase and GST Function in Response to GABA Induction of Melon Seedlings under Hypoxic Stress" by Li et al.
The paper is concisely written, however, several descriptions lack detail. Especially regarding the methods but also the breadth of discussions. Please find below some important suggestions that I would like the authors to address.

(1) More details on the growth conditions are needed: which lihght regime? Temperature? Humidity?

(2) The phylogenetic tree needs to be redone using a modern version of MEGA, like MEGA X. The authors need to: (i) use maximum likelihood methods with (ii) at least 100 bootstrap replicates; further they need to (iii) determine the best model for protein evolution by testing the models.

(3) Please broaden the scope of your analyses for the alignment. 
Indeed, various specialized metabolic routes occur across green algae and plants.
See and cite:
Rieseberg TP, Dadras A, Fürst-Jansen JMR, Dhabalia Ashok A, Darienko T, de Vries S, Irisarri I, de Vries J. 2022. Crossroads in the evolution of plant specialized metabolism. Seminars in Cell & Developmental Biology: S1084952122000738.

(4) The authors need to better explain their statistical analysis. How was this test determined as the best procedure?
Please test for normality of the data!

(5) Regarding figures 4 and 5: Please add pictures of how the plants looked like under these treatments. 
Where there any notable phenotypic changes? 

(6) The usage of GABA is found across various plants and algae, please read and cite "Heat stress response in the closest algal relatives of land plants reveals conserved stress signaling circuits" doi: 10.1111/tpj.14782 -- Plant J 2020 Aug;103(3):1025-1048
As well as
Kinnersley, A.M. and Turano, F.J. (2000) Gamma aminobutyric acid (GABA) and plant responses to stress. Crit. Rev. Plant Sci. 19, 479–509.
And
Li, Z., Yu, Z., Peng, Z. and Huang, B. (2016) Metabolic pathways regulated by c-aminobutyric acid (GABA) contributing to heat tolerance in creeping bentgrass (Agrostis stolonifera). Sci. Rep. 6, 30338.
And
The aldehyde dehydrogenase (ALDH) gene superfamily of the moss Physcomitrella patens and the algae Chlamydomonas reinhardtii and Ostreococcus tauri by AJ Wood, RJ Duff - The Bryologist, 2009 
This should be noted and discussed!

Reviewer 2 Report

The MS focused on changes in gene expression of GST due to hypoxic stress and GABA treatment. Hypoxic stress cause the oxidative stress in different plant parts such as root, stem and leaves. To cope with oxidative stress, amount and activities of antioxidants are generally increased. However, sometime level of anioxidants are low enought to protect membrane and cellular structures from oxidative stress. Exogenous GABA application stimulate antioxidant activity or increase the production of GST enzymes. The idea to work with is nice. However, I found several flaws in it to justify its publication. 

1. Results: The authors did not present the data of plant growth so that one can assess the extent of adverse effects of hypoxic stress on plant growth and any ameliorative effect of GABA in this regard.
2. Discussion part is not up to the mark, particularly first two paragraphs. The authors provide the repeated information of GST classes, and their functions. The results were only confirmed by citing previous studies. The Discussion should be interactive and inferential - what is outcome? 
3. Expression of GST was higher in leaves than in roots and stem, but activity of GST was greater in roots rather than in leaves. Over-expression of GST in leaves with low GST activity might have negative impact on cellular metabolism. How the authors justify this? These results were not discussed in Discussion part.
Page 1 Line 34 .... which are sent to vacuoles or plasmids[1]. 
Please replace "plasmids" with "plastids". 
It is suggestd that working hypothesis should be included in INTRODUCTION section. 
Discussion part needs to be revised.

Reviewer 3 Report

The article entitled "Molecular Cloning of a TCHQD Class Glutathione S-Transferase and GST Function In Response to GABA Induction of Melon Seedlings under Hypoxic Stress". The article was revealed that the full-length sequence of the CDS region of the CmGST gene was predicted via bioinformatics. Further, they have assessed the spatial expression of the CmGST gene, enzyme activity and GSH substrates induced in response to GABA at different treatment time and in different tissues under root hypoxic stress.

The manuscript comprises all the necessary elements of scientific paper. The manuscript needs considerable English correction. I recommend this article for publication after incorporating major changes given in below.

Authors have mentioned the cloning but the manuscript does not have any vector details. Please check it and add the same in the manuscript.

In materials and method section: authors should explain why each item of methodology was done.

The methods section should contain a framework figure of listing all the analysis/steps done in this paper.

Real-time fluorescence quantitative PCR section authors mentioned that they have designed the primer for obtained GST sequence. Please mention the primer designing tool in the section.

Please describe more about in the RNA extraction section in the materials and method section.

Line 112: Repetitive, so reframe it.

Line 157: p value should be p-value.

Fig 1. That can be moved to supplementary section.

If possible Phylogenetic tree should be differentiated by color.

Authors must concentrate on the formatting, and use of symbols, etc., in throughout manuscript.

Gene name should be in italics.

Organisms scientific name first mention only in full form all other mentions should be abbreviated. Example Cucumis sativus many places are in full form.

Authors should provide single space between manuscript lines and reference numbers in the manuscript.

Conclusion section looks shallow and write few lines about the future perspectives or hypothesize about the study. Discuss more and it will be useful to the readers for ease of understanding.

Round 2

Reviewer 3 Report

Authors carried out all my suggestions and cleared my queries. Further the manuscript is in sound scientifically. Therefore, I recommend the manuscript can be accepted for publication in its current form.

This manuscript is a resubmission of an earlier submission. The following is a list of the peer review reports and author responses from that submission.

Round 1

Reviewer 1 Report

In their manuscript, Li and colleagues report the cloning of a gluthatione S-transferase isoform from Cucumis melo (CmGST) and evaluated for the first time the impact of hypoxic treatments on its expression and total GST activity in melon. The study is motivated by the interest the topic bears from the horticultural point of view, given that hypoxic stresses can cause production losses in a relevant sensitive species such as melon, therefore deeper understanding of the ability of this species in ROS scavenging through glutathione would be desirable. However, the relevance of the results presented here is not clear. The study reports the expression pattern of one hypoxia-inducible GST gene from melon in response to the stress and exogenous GABA supply, but fails to discuss the significance of this observation in relationship with the hypothetical GABA metabolism in this species. The same applies to the described patterns of GST activity and GSH levels, whose significance is not properly discussed.

The literature provided about GABA and GSH metabolism under hypoxia in plants is largely incomplete for the scopes of the study. Furthermore, several citations in the reference list are related to studies published on non-indexed journals, which decreases the reliability of the information provided. In my check, references no. 10, 11, 12, 19,20, 22 and 24 would fall in this category, accounting for 28% of the overall literature cited. Moreover, most of these references consist in papers in Chinese language, which can be hardly accessible to the broader audience of foreigner readers (reff. 10, 11, 12, 19, 24).

The following points I made are specific for the core results of the study.

  • The cloning procedure of the novel sequence named as CmGST is not completely clear to me. It is stated that one couple of primers was designed “using the conserved sequence of GST as a reference” (p. 3 l. 118): has this information been extracted from an annotated genomic database existing for melo, or has it been reconstructed through homology analyses? In the second case, which GST sequence was used as the query? Which are the IDs of the hits predicted to encode for GST genes upon such a search, or their genomic locations on the melon genome assembly in the other case? Furthermore, GST genes usually belong to multigene families (as also pointed out in the introduction), therefore a PCR using primers designed on “conserved sequences” should be likely to return more than one amplification product, while it seems that only CmGST was cloned as a target: could the Authors comment on this aspect? Overall, it is unclear whether CmGST represents a new putative GST isoform that had not been previously annotated in melon (then why did the Authors opt for not making it available on public repositories?), or rather it is a predicted GST sequence that has now been verified by cloning (then the sequence should be referenced and the interest in cloning it should be clarified). Not all sequences used in Fig. 2 and Fig. 3 are referenced either.
  • Supplementary materials are mentioned in the text, but, to the best of my knowledge, such material cannot be found on the submission page. I was therefore unable to evaluate the statements regarding the predicted features of CmGST based on its primary sequence (p. 5, l. 177-184). Moreover, at p. 5 l. 180 the Authors provide specific information of the subcellular distribution of CmGST, but it is not clearly stated that this comes from in silico forecasts again, rather than from, for instance, microscopic imaging of a tagged version of the protein and signal quantification at a subcellular level. “The protein was predicted to be distributed…” would therefore be a clearer option for the text, in this occasion.
  • Section 3.3 has a wrong title.
  • The stress applied to seedlings should be more properly defined as “root hypoxia”, since roots were kept in oxygen-depleted media, while shoots were kept under aerated conditions. Fig. 4-6 show clear systemic effects of this kind of treatment, which should be better addressed in the manuscript.